# A Compilation of the Diverse miRNA Functions in *Caenorhabditis elegans* and *Drosophila melanogaster* Development

**DOI:** 10.3390/ijms24086963

**Published:** 2023-04-09

**Authors:** Daniel C. Quesnelle, William G. Bendena, Ian D. Chin-Sang

**Affiliations:** Department of Biology, Queen’s University, Kingston, ON K7L 3N6, Canada; d.quesnelle@queensu.ca (D.C.Q.); bendenaw@queensu.ca (W.G.B.)

**Keywords:** miRNA, non-coding RNA, *C. elegans*, *Drosophila*, development, gene regulation

## Abstract

MicroRNAs are critical regulators of post-transcriptional gene expression in a wide range of taxa, including invertebrates, mammals, and plants. Since their discovery in the nematode, *Caenorhabditis elegans,* miRNA research has exploded, and they are being identified in almost every facet of development. Invertebrate model organisms, particularly *C. elegans,* and *Drosophila melanogaster*, are ideal systems for studying miRNA function, and the roles of many miRNAs are known in these animals. In this review, we compiled the functions of many of the miRNAs that are involved in the development of these invertebrate model species. We examine how gene regulation by miRNAs shapes both embryonic and larval development and show that, although many different aspects of development are regulated, several trends are apparent in the nature of their regulation.

## 1. Introduction

MicroRNAs (miRNAs) have emerged as critical post-transcriptional regulators of gene expression [1,2]. This genre of regulators was first identified in the 1990s when Victor Ambros’s group was investigating the role of the RNA-binding protein *lin-4* in *Caenorhabditis elegans*, upon which it was discovered that *lin-4* was not a protein at all but rather a short RNA molecule with RNA-binding capabilities [3,4]. Not too long after, a second miRNA was identified in the worm by Gary Ruvkun’s group. This miRNA, called *let-7*, plays a crucial role in coordinating the timing of developmental events in the late larval stage of the worm [5]. miRNA research has since exploded, expanding to *Drosophila* and other invertebrate and vertebrate systems, including mammals, and is currently being conducted for therapeutic purposes [6,7,8,9,10]. According to the miRBase miRNA database, there are 253 known miRNAs in worms, 258 in flies, and over one thousand in mice and humans, and while several have been researched in great depth, the roles of most of them remain elusive [11,12,13,14,15,16]. Despite the advances made in mammals, it can be argued that *C. elegans* remains the champion of miRNA research due to its ease of use as a model organism [17]. In this review, we will explore the various roles miRNAs and miRNA families have throughout development in the invertebrate model organisms *C. elegans* and *Drosophila melanogaster*. We will be breaking this down at each critical stage in development, from events early in embryogenesis to larval development, highlighting the functions miRNAs play to shape the path for proper invertebrate development.

## 2. miRNA Biogenesis and Processing

The journey from miRNA transcription to activation is highly regulated, a necessity to ensure that both the correct miRNAs are created but also in the exact amount required for their specific functions [18,19]. The regulation itself justifies its own review and thus will not be covered in great detail here. Many reviews have covered miRNA biogenesis in great depth, including the following: [18,19,20]. Additionally, the mechanisms of biogenesis are greatly conserved, and as a result, invertebrate-specific components in this process are few.

miRNA sequences are often nestled in introns but can also be transcribed from exonic regions individually or in clusters of several miRNAs [21]. miRNA transcription is directed by RNA polymerase II (RNAPII) and occurs similarly to messenger RNA transcription [22,23]. Transcription by RNAPII produces an unprocessed pri-miRNA precursor containing a central hairpin surrounded by more than 1 kb of noncoding sequence [20,21]. The first processing step is carried out by an endonuclease complex called Microprocessor. Microprocessor is a heterotrimeric protein complex comprised of the endonuclease *Drosha*/*drosha*/*drsh-1* (gene names here are listed as human/fly/worm) and two molecules of *DGCR-8*/*pasha*/*pash-1,* which serve as co-factors. The microprocessor cuts the pri-miRNA transcript at the base of the hairpin and releases a shorter (roughly 60 nucleotides) miRNA precursor called the pre-miRNA [20,24,25,26]. The nucleocytoplasmic transport factor Exportin 5, along with its essential Ran-GTP co-factor, binds to the pre-miRNA and shuttles it out of the nucleus [18,27,28,29]. The next processing event is carried out by another endonuclease, *Dicer*/*Dcr-1/2*/*dcr-1*, which binds and cuts the pre-miRNA near the loop structure [30,31,32,33,34]. Cleavage of loop structure by Dicer releases a double-stranded RNA duplex which is then loaded onto an argonaute protein with the help of chaperone proteins *Hsc70*/*Hsc*-70/*hsp-1* and *Hsp90*/*Hsp-90/daf-21* [35,36,37]. There are 27 argonaute genes in *C. elegans*, though only several have been implicated in miRNA-mediated repression, most notably *alg-1* and *alg-2* [38,39,40]. In the fly, only one argonaute cooperates with miRNAs, *Ago1* [41,42]. The newly-formed complex of the argonaute and miRNA duplex, often termed the pre-RISC, hastily removes the passenger strand of the miRNA. While both strands are capable of functioning as mature miRNAs, the orientation at which the miRNA duplex enters the argonaute decides which strand ultimately becomes the active strand and which is discarded [43,44,45]. Generally, the strand with the least stable 5’ terminus is selected as the active strand [44]. Another factor of many that determine the selection of the active strand is the presence of a U in the first position of the strand [46], although this is considered to be more of a guideline for strand selection. Upon strand selection and passenger strand release, the argonaute-miRNA complex now considered a loaded RISC or miRISC, can carry out its function of inhibiting translation by its binding to messenger RNA. A schematic of the steps in miRNA processing can be found in Figure 1A.

## 3. miRNA Function and Decay

Mature miRNAs that are loaded in the RISC complex must then be taken to their target messenger RNAs so that they can carry out their regulatory function. More specifically, miRNAs bind to specific target sequences, termed seed sequences, located in the 3′ UTR of these transcripts [18,19]. These sequences are generally seven bases in length and align with bases 2–8 in the miRNA [18,47]. Many genes are thought to have conserved miRNA seed sequences in their 3′ UTRs (60% of genes in humans, less in invertebrates), and these estimates do not account for seed sequences for non-conserved miRNAs [47]. Additionally, miRNAs are classed into families based on the sequence similarity of their seed sequences [18,47].

Binding of a miRNA to its seed sequence in the 3′ UTR of a messenger RNA induces a cascade of protein recruitments that lead to either the inhibition of translation or the degradation of that transcript (Figure 1B). Immediately following miRISC binding to the seed sequence, the adaptor protein GW182/AIN-1 (fly/worm, commonly referred to as GW182) is recruited [48,49]. GW182 then recruits the deadenylases and poly-A binding proteins CCR4-NOT, PAN2, PAN3, and PAB-1 which are required for inhibiting translation and/or degrading the transcript [49,50,51,52]. The mechanism of translational repression is understood to be a result of the miRISC-recruited machinery interfering with translation initiation factors, mainly the 5′cap binding heterotrimeric protein complex eIF4F, thus preventing the initiation of translation [53,54]. CCR4-NOT and PAN2-PAN3 both have the capability to deadenylate transcripts and in this process, they mediate the deadenylation of the poly-A tail, thus destabilizing the transcript and triggering its degradation [19,51,52]. GW182 and poly-A-binding proteins such as PAB-1 tether these complexes to the RISC and the transcript, respectively [49,50]. Together, the miRISC, GW182, and other recruited factors downregulate mRNA expression by translational repression, transcript destabilization, and degradation or both. It is important to note here is that although these mechanisms are generally well-understood, the mechanism for how the miRNA is transported from the site of its binding to the RISC complex to the site of translation of its target transcript remains unclear.

Equally as important as the biogenesis of miRNAs is their turnover or decay. The abundance of a miRNA is important for the regulation of mRNA targets and, when misexpressed, can negatively lead to improper changes in their expression [20]. The exact mechanisms for miRNA decay are not fully understood, but it is believed that the cell adopts several strategies for destabilizing and clearing miRNAs (Figure 1C). First, a miRNA-decay complex has been identified consisting of the exoribonuclease XRN-2 and the de-capping scavenger protein DCS-1, which can degrade many miRNAs, including *let-7* [55,56]. Another mechanism for miRNA degradation is the addition of several nucleotides at the 3′ end of the miRNA by poly-A polymerases such as Wispy, thus destabilizing and clearing many miRNAs at once [57,58]. miRNAs can also be cleaved by a component of the RISC complex, Tudor-SN, at specific sites within their sequence [59,60]. In a more recent study, a mechanism for miRNA decay was uncovered that is seed-sequence dependent, in which the seed sequence of the miRNA was crucial in recruiting the ubiquitin ligase TDMD/EBAX-1, ultimately leading to its decay. EBAX-1 is thought to be recruited either by the recruitment of an RNA-binding protein first or by conformational changes in the argonaute, but the exact mechanism remains unclear [61,62]. Taken together, the mechanisms for miRNA degradation are crucial for maintaining proper miRNA levels but appear to be conditional.

## 4. miRNAs in Embryonic Development

### 4.1. Maternal-to-Zygotic Transition

miRNAs are present at the earliest stages of embryo development. The first critical step in invertebrate embryonic development is the transition from the use of maternal genetic material to that produced by the zygote, termed the maternal-to-zygotic transition, or MZT [63,64,65]. Detection of zygotic transcription in *C. elegans* can be detected as early as the 4-cell stage and is exponentially greater 90 min thereafter. *Drosophila* zygotic expression occurs within the first 2.5 hours post-fertilization [66]. A key process during MZT is the clearance of maternal messenger RNA to allow for the exclusive expression of the newly-transcribed zygotic mRNA [66,67,68]. In *Drosophila* this process occurs 1–2 h into embryonic development and is largely driven by the expression of the miRNA *miR-309* [69,70]. *miR-309* clears maternal mRNAs by direct binding to its seed sequence in their 3′ UTRs, resulting in the degradation of its targets [69,70]. *miR-309,* along with many other early zygotic miRNAs, are activated by the RNA-binding protein Smaug (SMG) [71]. SMG mutants exhibit a delay in the production of zygotic miRNAs, and maternal miRNAs that are targets of *miR-309* and other miRNAs at this stage are unable to be cleared [71]. SMG, along with other RNA-binding proteins, Brain Tumor (BRAT) and Pumilio (PUM), clear maternal mRNAs through direct binding, although not redundantly with miRNA-mediated clearance [71,72]. *miR-309* also experiences levels of spatial regulation. The *miR-309* cluster is one of eleven clusters activated by the zinc-finger protein Zelda through enhancer binding, and Zelda mutants display *miR-309* expression only in the anterior half of the embryo [73]. In Bicoid mutants, *miR-309* is expressed in the anterior half early, but then that expression is lost as development proceeds, suggesting spatial co-regulation between Bicoid and Zelda [73].

The role of *Drosophila miR-309* in maternal mRNA clearance is functionally similar to that of vertebrate *miR-430*, suggesting a conserved maternal mRNA degradation pathway [74,75,76]. However, miRNAs are not believed to act in the MZT in *C. elegans* [77]. Rather, *C. elegans* maternal product clearance is thought to be directed by small noncoding RNA molecules called 22G-RNAs that are activated by the argonaute CSR-1, both of which are maternally inherited [77,78]. While small RNAs are essential for the clearance of maternal transcripts in flies and worms, the mechanisms differ slightly, and many aspects of this process are still unclear.

Early embryos can also contain miRNAs that are inherited maternally [79]. This is more apparent in *C. elegans* as it is unclear whether maternal miRNAs are found in *Drosophila* embryos. The *C. elegans mir-35*, family of miRNAs, are among those expressed maternally in the early embryo [79,80]. The *mir-35* cluster encodes 8 miRNAs, *mir-35-42* [81,82]. *Mir-35* family members are also expressed at the start of zygotic transcription and are strongly expressed until the mid-stage embryo. Transcription of *mir-35* declines after gastrulation (350 nuclei) [80]. In addition, either version of the miRNA (maternal or zygotic) is sufficient for embryonic development, although it has been hypothesized that both are required for its functions later in development and, without one or the other, there would not be enough *mir-35-42* accumulation [79,82]. Evidence suggests that the *mir-35* family delays the expression of sex-specific genes so that sex determination doesn’t occur too early, thus acting similar to a developmental timer [79]. It does this by regulating the translation of two downstream genes that promote sex determination, *sup-26* and *nhl-2* [83,84]. *Sup-26* and *nhl-2* regulate an array of genes, including the master regulator gene for male development *her-1* [83,85,86]. By repressing genes that promote sex-determining decision-making, the *mir-35* family is able to keep the embryo in a state of undifferentiated sex [79].

### 4.2. Programmed Cell Death (PCD)

The *mir-35* family of miRNAs have a plethora of other roles in *C. elegans* development post-gastrulation. The expression of this family of miRNAs ranges throughout embryogenesis and after hatching. Using GFP transcriptional reporters, *mir-35* is expressed in most cells from the early embryo until the late embryo elongation stage, and a member of the *mir-35* cluster *mir-42* is expressed in the hypodermis during and after embryogenesis [82]. Additionally, a knockout of the entire family results in embryonic lethality, suggesting essential roles for this family in the embryo [82]. Following gastrulation *mir-35-42* is involved in controlling apoptosis timing for cells that are programmed to die. As cells divide in the embryo, the mothers of those cells die when they are no longer needed [87]. The *mir-35* family of miRNAs plays a central role in this process by regulating the expression of genes required for apoptosis [88]. One such gene is *egl-1,* which, when expressed at high levels in mother cells targeted for apoptosis, will promote cell death [89,90]. *Mir-35* suppresses *egl-1* to maintain low levels of *egl-1* transcript abundance in PCD-targeted mother cells to prevent premature cell death by mediating the deadenylation of *egl-1* [88]. This regulation is found in germ cells as well, as *mir-35-42* inhibits *egl-1* and another gene, *nck-1*, to prevent apoptosis in the germline [91].

The repression of *egl-1* in the embryo is cooperative, with *mir-35* coordinating with another miRNA *mir-58* to control *egl-1* levels [88]. *Mir-58* is the homolog of the *Drosophila bantam* miRNA, which also functions within the realm of embryonic apoptosis. In flies, *bantam* prevents apoptosis by regulating an activator of apoptosis, *head involution defective* (*hid),* by translational inhibition [92]. *Hid*, along with *reaper* (*rpr*) and other *reaper*-family genes, promotes apoptosis directly by preventing caspase inhibition and indirectly by facilitating the degradation of apoptosis inhibitors (IAPs, inhibitors of apoptosis proteins) [93]. *Hid* and *rpr* expression is regulated by another miRNA, *miR-11,* in early and mid-stage fly embryos [73,94]. *Rpr* was shown to be overexpressed in flies with decreased *miR-11* function, and in these same animals, *hid* was prematurely expressed before gastrulation, an abnormality as *hid* is not expressed until after gastrulation in wild-type animals [73]. This data suggests a role for *miR-11* in directly repressing *hid* and *rpr* and inhibiting precocious cell death. Misexpression of pro-apoptotic genes such as *egl-1* or *hid* in embryonic cells targeted for cell death results in premature apoptosis, thus the importance of miRNAs to ensure that the correct timing of cell death is carried out.

### 4.3. Mid-Late Embryonic miRNAs

In worms, another major family of miRNAs is the *mir-51* cluster, consisting of 6 members, *mir-51-56* [95]. Together with the *mir-35* family, they encompass nearly 50 percent of miRNAs present during embryogenesis [96,97]. *Mir-51* family members belong to the highly conserved *mir-100* family of miRNAs and, as observed in *mir-35* family mutants, exhibit high levels of redundancy [82,95,96]. Knockouts of the entire *mir-51* family result in embryonic lethality that can be rescued by reintroducing any one member, and knockouts of individual members produce no obvious phenotypes [82,95]. Although many of the aforementioned embryonic miRNAs are important but not essential for development, *mir-51* is one of two miRNAs that are (the other being *mir-35*), suggesting crucial roles for this family in embryogenesis [97].

Several functions for this family have been identified in *C. elegans* mid-late embryonic development. One such role is involvement in the attachment of the pharynx to the anterior hypodermis, in which *mir-51* expressed in the arcade cells regulates the fat cadherin ortholog *cdh-3* [96]. Overexpression of *cdh-3* in *mir-51* mutants enhances the unattached pharynx (*pun*) phenotype [96]. Along with defects in pharyngeal attachment, *mir-51* family mutants exhibit abnormal epidermal morphology [82,96,97], although the mechanisms underlying these defects remain unclear. *Mir-51* also contributes to neuronal morphology by promoting Gamma-aminobutyric acid (GABA)ergic synapse formation and GABA receptor abundance [98]. This is done through the repression of the lysosomal trafficking-related Rab guanine-nucleotide exchange factor (GEF) *glo-4* in GABAergic motor neurons by upregulating the Rab *glo-1* and the lysosomal cargo sorting gene *ap-3*, increasing the number of synapses and the amount of GABA receptor, respectively [98]. These are two of many possible roles for this ancient family of miRNAs in the worm. miRNA target prediction algorithms such as TargetScan and PicTar return lists of hundreds of putative targets for this family [99,100,101]. Moreover, none of the aforementioned roles for *mir-51* appear to explain the embryonic lethality observed in *mir-51* family knockouts [97], implying that the ‘main’ role of the family remains elusive.

## 5. miRNAs in Larval Development

### 5.1. Developmental Timing

While it is clear that miRNAs are essential for embryonic development, their roles in post-embryonic development are more greatly understood. The first miRNAs discovered (*lin-4* and *let-7* in *C. elegans*) play key roles in larval development and have been a central focus in the miRNA field since their discovery. We know now of a plethora of miRNAs controlling pathways relating to developmental timing, tissue-specific growth, cell-fate determination, and others all throughout the invertebrate larvae [102,103].

Perhaps the most well-studied role for miRNAs across all species is that of regulating developmental timing in *C. elegans*. This role lies within a pathway known to worm researchers as the heterochronic pathway, in which the timing of developmental events, mainly somatic cell divisions, are regulated [104]. This pathway has been at the center of miRNA research since its discovery and has been reviewed in great depth [103,105,106]. There are four larval stages in the worm, characterized by cell divisions into stage-specific lineages, and the timings of these divisions at each larval stage are regulated. Some of the genes responsible for controlling developmental timing are *lin-14*, *lin-28,* and *lin-29*, and were first identified due to their phenotypes showing delayed and repeated developmental events while other events proceeded as normal [106]. These defects have been observed at all larval molts in the worm providing evidence that the timing of these developmental events is tightly regulated throughout larval development [103]. miRNAs play a central role in this process, regulating the expression of genes in order to ensure the correct timing of these events. In the context of heterochronic miRNAs, the pathway can be split into two parts: regulation of developmental timing in the early larval stages and in the late larval stages. Each part is regulated by one miRNA: *lin-4* controls early larval timing, and *let-7* controls late larval timing (Figure 2A) [103,106].

The *lin-4* miRNA regulates developmental timing at the L1-L2 transition by directly inhibiting *lin-14* through binding to its 3′ UTR [103,106]. *Lin-14* promotes L1 cell lineages and must be inactivated to proceed past the L1 stage. Failure to do so results in the reiteration of L1 cell fates and prevents the worm from proceeding with L2 lineages [103,106]. These reiterative phenotypes are present in both *lin-14* overexpression and *lin-4* loss-of-function mutants, providing evidence for the opposing roles they play in this pathway [103,112]. Conversely, *lin-14* loss-of-function mutants skip the L1 cell lineages and proceed immediately with L2 lineages [103,112]. *Lin-14* also controls the timing of cell-specific morphological changes, particularly in several motor neurons which remodel their synapses at the end of the L1 stage [106].

This form of regulation by *lin-4* occurs again in a similar fashion to promote L3 cell fates at the L2/L3 transition. In this iteration of heterochronic regulation, *lin-4* targets the L2 lineage-promoting gene *lin-28,* which, when not repressed, causes repeated L2 cell fates [103,106,107]. *Lin-4*-mediated repression of *lin-28* occurs upstream of genes L3-promoting *lin-46* and L2 promoting *hbl-l* [103,107]. Additionally, *hbl-1* is regulated by *let-7* family members *mir-48/84/241* at this stage [113]. *Lin-28*, as well as its role as an L2 lineage promoter, also regulates *let-7*, preventing its precocious expression [107]. In this way, *lin-28* is bridging the gap between early- and late-larval heterochronic regulation.

The second half of larval developmental timing is controlled by *let-7*. The *let-7* family of miRNAs is highly conserved across species, with *C. elegans* coding for nine members [105]. *Let-7* expression begins at low levels around the L2-L3 molt and increases into the late L3 and L4 stages [105]. It has many targets, but in this particular pathway, it represses two genes, *lin-41* and *hbl-1*, to promote adult cell fates [106]. *Lin-41* and *hbl-1* inhibit the *lin-29* transcription factor at the translational and transcriptional levels, respectively, and by doing so, promote L4 fates [103,106]. One additional factor in this pathway is that HBL-1, when expressed, inhibits *let-7* in a negative feedback loop, which in itself prevents *let-7* from being precociously expressed [114]. As observed with *lin-4/lin-14*, *let-7* and *lin-41/hbl-1* mutants have opposing phenotypes, displaying either repeated cell fates or lack thereof, depending on the mutation [103,106]. *Let-7*-mediated repression of these two targets therefore allows *lin-29* expression and the subsequent L4-adult transition to occur naturally. In a separate pathway, *let-7*-mediated regulation of *lin-41* in the vulval-uterine system is necessary for proper vulval development and the prevention of vulval bursting that is observed in *let-7* mutants [115].

Contrary to the large *let-7* family in worms, *Drosophila* only contains one *let-7* miRNA [105]. Similar to its homolog in the worm, *Drosophila let-7* has roles in developmental timing. *Let-7* is transcribed from a cluster also containing the *lin-4* ortholog *mir-125* and *mir-100* and is expressed in late instar larvae, around the 3^rd^ larval stage [105,116]. It was found to be involved in controlling the timing of abdominal neuromusculature junction remodeling [117]. The metamorphic stage of fly development sees the remodeling of its internal tissues, and the timing of this remodeling is crucial for proper development [118]. *Let-7* mutants appear to display juvenile-like features in their neuromusculature junctions, a trait that is analogous to the reiterative larval cell lineages in the worm [117]. In this pathway, *let-7* binds to and suppresses *abrupt*, a gene required ubiquitously throughout development, down to nearly undetectable levels by the early adult stages [119]. The downregulation of *abrupt* in abdominal-muscle cells is a contributor to promoting the transition to adult morphology of the abdominal neuromusculature. The regulation of abrupt is not the only heterochronic role for *let-7* in flies. *Let-7* plays a role in specifying the fates of mushroom-body (MB) cells, a pair of neurons involved in olfactory learning and memory [120]. During the larval-to-pupal transition, neuronally-expressed *let-7* downregulates *chinmo* (Chronologically Inappropriate Morphogenesis) by direct binding to sites in its 3′ UTR [121]. In mutants with insufficient or too much *let-7*, the MB neurons experience changes in the timing of their cell fate transformations [121]. *Let-7* regulation of *chinmo* has also been shown to have effects on lifespan in adult flies as well [122]. Altogether, *let-7* has crucial heterochronic roles that are conserved throughout invertebrate models.

Curiously, the *C. elegans mir-51* family of miRNAs have roles in or related to this pathway as well. Loss of *mir-52* rescues the *mir-48/84/241* heterochronic phenotypes but in turn, also causes an additional L4-adult molt [123]. At the L2/L3 molt, loss of *mir-52* also rescues *lin-46* mutant phenotypes where the L2 larval lineages are reiterated [123]. Similar phenotypes are observed at the L2/L3 molt in animals lacking the entire *mir-51* family [123]. The precise role of *mir-51* family members in this pathway is not yet clear, but it is evident that developmental timing is one of, if not the most prominent functions of invertebrate miRNAs.

### 5.2. Neuronal Asymmetry

Cells arising from the same lineage may experience changes in their individual fates resulting from suites of gene expression changes. This is particularly evident in the *C. elegans* nervous system, as it gives rise to many pairs of neurons. In the ASEL/ASER pair of head neurons, the expression of one miRNA or another is sufficient in determining their cell identities and functions as separate taste receptors [109]. In this instance of left/right asymmetry, two miRNAs are responsible for regulating the expression of genes that promote one cell fate or the other [124]. In ASEL, the *lsy-6* miRNA promotes the left cell fate by repressing a transcription factor called *cog-1* [110]. In ASER, *lsy-6* is inhibited by an upstream miRNA called *mir-273* [111]. *Mir-273* suppression of *lsy-6* occurs in a regulatory cascade through the transcription factor *die-1*, which promotes *lsy-6* expression (Figure 2B) [109,111]. In ASEL *cog-1* is not inhibited by *lsy-6* and downregulates the *lim-6* transcription factor, which in turn inhibits the expression of the guanylyl cyclase *gcy-5* [109]. *Gcy-5* is one of three *gcy* genes expressed by the ASE neuron pair along with *gcy-6* and *gcy-7* [109,111]. For proper chemosensation, ASEL must express *gcy-6* and *gcy-7* but not *gcy-5*, while ASER must express only *gcy-5*, and improper *gcy* expression in these cells result in worms that are unable to distinguish between different chemicals [109]. This system illustrates how the miRNAs *lsy-6* and *mir-273* are crucial for controlling the functions of individual neurons in the worm.

### 5.3. Muscle Development

In addition to specifying individual cell fates and their timing, miRNAs have also been discovered to have roles in tissue development. To understand how this is occurring, the prime example is the role of *mir-1* in the developing muscle tissue. Both the *miR-1* (*mir-1* in the worm) sequence and its function in the developing muscle tissue are highly conserved in flies, worms, and mammals [124,125,126]. In *Drosophila miR-1* is expressed in mesodermal cells and, by the larval stages, many muscle cells, including somatic, visceral, and pharyngeal muscle cells [124]. The same study by Sokol and Ambros shows a role for *miR-1* in muscle tissue downstream of transcription factors *twist* and *Mef2*, in which mutants lacking *miR-1* display deformed musculature and are unable to survive through the larval stages [124]. The study concludes that the role of *miR-1* in the muscle was not to do with muscle formation but rather its maintenance and growth during the larval stages [124].

The role of *miR-1* in the developing fly muscle is consistent with studies of *mir-1* in *C. elegans* which have shown that *mir-1* is essential for maintaining muscle physiology. *Mir-1* mutants in the worm exhibit a spectrum of muscular defects, including mitochondrial fragmentation, decreases in cellular ATP, and cell–cell fusion defects at the L1 larval stage [125]. These defects are believed to be a product of these cells’ inability to acidify their lysosomes due to defects in the formation of V-ATPase complexes. In fact, 15 V-ATPase genes have *mir-1* binding sites in their 3′ UTRs, and it is likely that *mir-1*-mediated regulation of these subunits is the cause of the muscle defects observed in *mir-1* mutants. The V-ATPase complex requires specific concentrations of each of its subunits to properly form; thus, it is plausible that *mir-1* is responsible for modulating the levels of each subunit, ensuring they are neither over- nor under-expressed [125,127]. Another functional target of *mir-1* is the mitochondrial turnover- and homeostasis-controlling gene *dct-1*, whose role in this process, although not yet clear, is thought to be related to the mitochondria defects observed in *mir-1* mutants [125]. Additionally, *mir-1*-mediated regulation of the muscle-specific transcription factor *mef-2* has been shown to have effects on the formation and signaling between neuromuscular junctions [128]. *Mef2* is also regulated by the miRNA *miR-92b* in a negative feedback loop to maintain stable levels of both components in *Drosophila*, indicating that the regulation of *Mef2* by miRNAs is conserved throughout invertebrate species [129]. Taken together, *mir-1* affects muscle development through the regulation of many targets and in both flies and worms, *miR-1*-mediated repression of *Mef2/mef-2* appears to be critical for the muscle to develop properly.

Another role for miRNAs related to muscle development is the suppression of muscle-promoting genes in *Drosophila* tendon cells. Tendon cells can express muscle genes when their fates are determined, and if they don’t suppress muscle gene expression, the ability to retain their identity is compromised. The miRNA *miR-9a* prevents muscle genes from being expressed, which retains the proper cell fates of tendon cells, a role important for proper muscle-tendon interactions [130]. In an additional and separate pathway, *miR-9a* was recently shown to regulate the zinc-finger transcription factor *senseless* in the developing peripheral nervous system (PNS). Senseless is required for PNS development and controls the specification of sensory organ precursor (SOP) cells in the early embryo [131]. In this role, *miR-9a* fine-tunes *senseless* expression to make sure the proper number of cells become SOP cells [131].

### 5.4. Homeotic Gene Regulation

The morphogenesis of vertebrate and invertebrate species is driven by segmental gene expression. In this process, the developing animal is divided into compartments/segments, each with their own suite of expressed genes that lead to the differentiation of tissues and body parts. These genes, termed homeotic, homeobox, or hox genes, are expressed spatially and temporally in their respective segments and are tightly regulated as misexpression of these genes can lead to the growth of alternative body parts [132,133,134]. Regulation of Hox genes comes in many forms, notably through the expression of other Hox genes [135]. It is now understood that Hox genes may be regulated by miRNAs as well. The system best showcasing the role of miRNAs in Hox gene regulation is the bithorax complex in *Drosophila*. The bithorax complex (BX-C) is a complex of Hox clusters that determine the fate of the third thoracic and the eight abdominal segments of the fly [136]. The cluster contains nine cis-regulatory regions: *abx, bxd, iab-2, iab-3, iab-4, iab-5, iab-6, iab-7, and iab-8* [137]. These are arranged along the chromosome in the same order as the parasegments that they act in. The bithorax complex encodes three homeobox genes, *Ultrabithorax* (*Ubx*), *Abdominal A* (*Abd-A*) and *Abdominal B* (*Abd-B*) [135]. Additionally, the whole BX-C region is extensively transcribed thus it is believed that it also encodes a number of regulatory RNA molecules [136]. The most well-known regulatory RNA produced in the BX-C is the miRNA *miR-iab-4*. *miR-iab-4* (or just *iab-4*) is transcribed from the *iab-4* cis-regulatory region, and encodes for a pair of miRNAs, one from each arm of the pre-miRNA hairpin, termed *iab-4-5p* and *iab-4-3p* (Figure 3) [138]. Both *iab-4-5p* and *iab-4-3p* target *Ubx* through direct binding to *iab-4* seed sequences in its 3′ UTR [135,138,139]. Ectopic expression of these miRNAs inhibits the accumulation of *Ubx,* and these animals go on to grow wings instead of halteres in the 3rd thoracic segment, a phenotype observed in *Ubx* loss-of-function mutants as well [138,140].

The antisense strand complementary to the *iab-4* locus contains a similar pre-miRNA hairpin structure that encodes a third miRNA initially termed *iab-4AS* but now known as *miR*-*iab-8*. The *miR*-*iab-8* miRNA is encoded in the 5^th^ intron of the *iab-8* long noncoding RNA (lncRNA), which is transcribed on the antisense strand in the distal-proximal direction [141]. As the *iab-8* lncRNA promoter is located in the *iab-8* cis-regulatory region, the resulting miRNA Is expressed mainly in abdominal segments expressing Abd-B. *miR*-*iab-8* has complementary sites in both *Ubx* and *Abd-A* and represses both by direct binding to their 3′ UTRs. Furthermore, expressing *miR*-*iab-8* in the haltere imaginal disc produces the same haltere-to-wing phenotype as mentioned earlier [140]. It is understood that *miR*-*iab-8* suppresses *Ubx* and *Abd-A* in the posterior abdominal segments, as an expression of those genes promotes anterior segment identities. Thus, even non-coding RNA genes such as *miR-iab-4* and *miR-iab-8,* which are expressed more posterior in the Bithorax Complex, are repressing anterior Hox genes, *Ubx* and *Abd-A,* and is consistent with previous knowledge that Hox genes generally regulate genes located more anteriorly than themselves, a phenomena known as the posterior dominance rule [142]. Interestingly, the proximity of the 3′ end of the *iab-8* lncRNA to the *Abd-A* promoter region suggests that *iab-8* transcription also interferes with the *Abd-A* promoter, creating a second, redundant level of *Abd-A* regulation [141].

Although *C. elegans* development is not driven by segmentation such as *Drosophila*, many Hox gene homologs in the worm are still required for defining spatial positioning [143]. This is partly due to the fact that Hox genes are not clustered in the same manner and are not expressed collinearly as they are in the fly [144]. Several Hox gene homologs still share similar expression patterns to their fly counterparts as well. For example, the worm homolog of *Abd-B*, *nob-1*, is expressed posteriorly similar to *Abd-B* in the fly, indicating some conservation of the expression of Hox genes in worms [145]. Interestingly, *nob-1* has been shown to be regulated by the miRNA *mir-57* early in development [146]. Animals overexpressing *mir-57* display morphological defects in their posterior region, often called variable abnormal or Vab phenotypes. These phenotypes are also observed in *nob-1* null mutants, suggesting that *nob-1* is regulated directly by *mir-57* and that it is functioning in a similar manner to *Abd* in the fly [146]. Reduction in *nob-1* also decreases *mir-57* expression and function, indicating that *nob-1* is also acting upstream to *mir-57* in a feedback loop where it positively regulates *mir-57* [146]. Hence, *mir-57*-mediated repression of *nob-1* is essential for posterior patterning in the worm.

### 5.5. miRNAs in the Developing Appendages

miRNAs participate in the regulation of the developing fly wing. The wing is formed from the wing imaginal disc at the first instar larval stage and then grows dramatically during the second and third instar larval stages, characterized by cell proliferations and identity changes [147]. There are several key genes in this process that must be expressed at the correct levels, notably *vestigial* (*vg*) in cells that become the wing and the morphogens *decapentaplegic* (*dpp*) and *wingless* (*wg*). Factors are required for cell identity specifications later in wing development. This includes the LIM-only factor *dLMO* [148,149,150,151]. Overexpression of *dLMO* in the wing results in a significant reduction in the wing margin, and a loss of *dLMO* produces a curly wing phenotype [152]. The *dLMO* transcript is regulated in its 3′ UTR by *miR-9a* [152]. Moreover, *miR-9a* mutants display the *dLMO* gain-of-function reduction in wing margin phenotype, suggesting opposite roles for the two genes [152]. The loss of wing margin is attributed to increased amounts of apoptosis in the dorsal wing primordium as a result of increased *dLMO* function [153]. For these reasons, it is understood that the role of *miR-9a* in this process is not to inhibit *dLMO* but rather to maintain its expression at a stable level while the wing develops.

Cells in the developing wing also express miRNAs *let-7, miR-125,* and *bantam*. For example, *let-7* and *miR-125* are thought to have a role in controlling the timing of cell-cycle exit in the wing, yet another example of their heterochronic capabilities [119]. *let-7, miR-125* double mutants display smaller wings than wild-type animals, and these wings contain more, smaller cells, the latter being expected from cells that are not able to exit the cell cycle [119]. Another miRNA expressed in the imaginal wing disc is *bantam*, which controls apoptosis through its regulation of *hid* as described earlier [92,154]. The role of *bantam* in the wing disc may be related to *miR-9a,* which indirectly regulates apoptosis through its regulation of *dLMO*. Taken together, complex processes such as *Drosophila* wing development require many miRNAs working in tandem to regulate the expression of their target genes.

The *Drosophila miR-309-6* miRNA cluster has a distinctive composition of miR-6/5/4/286/3/309, which is conserved across the genus. Expression of *miR-309-6* in *Distal-less* or *patched* expressing cells in the leg and wing discs results in the shortening and deletion of leg tarsal segments relative and loss of the wing anterior cross-vein. Overexpression of either *miR-309-5* or *miR-6-1/6-2/6-3* mimics the phenotype observed by overexpression of the entire cluster. Tarsal segment deformities include loss of segment, joint boundaries, and claws. *miR-309-5* and *miR-6-1/6-2/6-3* downregulate numerous genes known to be involved in either proximal-distal patterning of appendages such as *zfh-2*, a zinc finger homeodomain-2 transcription factor [155]. Transcription factors *Sp1* and *dysf* are also downregulated, which are required for appendage growth and tarsal joint formation in insects [156,157]. RNA levels for *Egfr* and *dpp,* known to be vital for leg patterning, are also decreased [158,159]. Together, the *miR-309-6* miRNA cluster plays a crucial role in the developing leg imaginal disc.

Although flies have developed small-RNA-mediated mechanisms to regulate appendage development, several of these mechanisms are not present in *C. elegans* as they do not have appendages. In some cases, the miRNA is not expressed in the worm; for example, both the *miR-309-6* and *miR-6* clusters of miRNAs that are required in the developing fly leg have no counterpart in the worm [16]. In other cases, the miRNA is expressed but functions in an entirely different process. Drosophila *miR-9a* is essential for wing development, but in *C. elegans*, the *miR-9a* homolog *mir-79* is expressed in the epidermis and regulates neuronal branching and migration non-autonomously [16,160]. These miRNAs are highly conserved but have gained separate roles as these species diverged [16].

## 6. Concluding Remarks

miRNAs are implicated ubiquitously throughout invertebrate development, starting from before gastrulation to the moments before the animals become adults. They are involved in a number of cellular processes, such as apoptosis, cell division, and morphogenesis. They also play crucial, conserved roles in tissue-specific development, such as the role of *miR-1* in the muscle, for example. miRNAs are being discovered to have roles in neuronal development that in itself warrants their own review (see Ref [7]). There are also many additional miRNAs that are essential for the development of invertebrates that were not touched upon in this review. Furthermore, miRNAs in vertebrates and mammal models have become a research field of great interest for their therapeutic capabilities [161,162]. Understanding the roles of miRNAs during development can contribute to our understanding of human diseases such as cancer [163,164]. *let-7*, for instance, is a known tumor suppressor that regulates the oncogene *Ras* [163,165]. Our knowledge of the roles of miRNAs in disease has grown enough to also warrant its own review. The aim of this review is to compile and highlight many of the main miRNA players in the development of the invertebrate model organisms *C. elegans* and *Drosophila melanogaster* (Figure 4).

Although the miRNAs discussed in this review have vastly different roles throughout development, many of them share similar trends in the nature of their regulation. The traditional view of the function of a miRNA is for it to prevent its target gene from being expressed, whether doing so by targeting it to be degraded or by blocking translational machinery to prevent the expression of its protein. However, many miRNAs don’t have that role and instead only modulate the amount of their targets to maintain a specific concentration of the target protein. This is evident looking at *miR-1* in muscle cells making sure the exact amount of each V-ATPase complex subunit, or *miR-9a* ensures that the correct amount of *dLMO* is expressed in the developing fly wing. In these examples, too much or too little expression of their target genes can have detrimental effects on their respective processes, such as failure of lysosomal acidification and subsequent defects in the whole muscle tissue. Given these detriments, it could be theorized that many (or maybe all) of the genes required during development have their expression tuned to specific levels by vast networks of miRNAs.

Another trend that can be observed has to do with controlling the timing of developmental processes and events. The immense roles of *lin-4* and *let-7* in controlling the timing of cell lineages and larval molts are classic examples of this role for miRNAs during development. However, this is not specific to the *C. elegans* heterochronic pathway but appears to be more of a general role in miRNA-mediated gene regulation. Examples of this include *mir-35* family regulating the timing of sex determination and apoptosis in early embryos and *let-7/miR-125* regulating the timing of cell-cycle exit in the wing imaginal disc. This inherently makes sense as developmental processes and events are highly coordinated and must all occur at the right time relative to each other. Furthermore, within these processes and events, genes cannot be expressed too early or too late, or the resulting embryos or larvae show defects and possibly lethality in these stages. Therefore, miRNAs are perfectly suited to ensure the exact timing and expression of developmental genes. Altogether, the roles of miRNAs during invertebrate development are vast and ever-growing, with many more being discovered and characterized at an exponential rate.

## Figures and Tables

**Figure 1 ijms-24-06963-f001:**
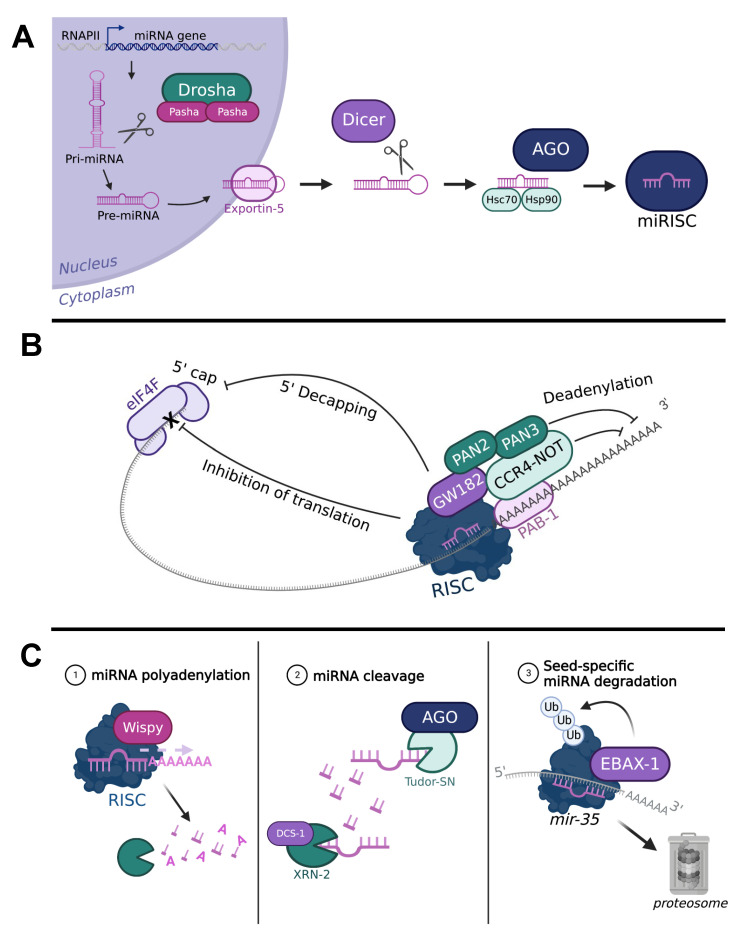
An overview of miRNA biogenesis, function, and decay. (**A**) The miRNA transcript undergoes several cleavage steps (scissors) by various endonucleases, Drosha/Pasha and Dicer. The miRNA is loaded onto an argonaute protein (AGO) with the help of chaperone proteins (Hsc70/Hsp90); after miRNA strand selection occurs, the miRISC complex is formed. (**B**) miRNA binding to the 3′ UTR of an mRNA recruits the adaptor protein GW182, followed by the deadenylases and poly-A binding proteins CCR4-NOT, PAN2, PAN3 and PAB-1. Recruitment of these proteins induces translational repression by interfering with 5′ cap-binding protein complex eIF4F, 5′ de-capping, or deadenylation in the 3′ poly-A region. Adapted from Ref. [19]. (**C**) miRNA turnover is induced by either the addition of several A nucleotides by poly-A polymerases (e.g., Wispy) at the 3′ end of the miRNA (left panel), cleavage by nucleases such as XRN-2 or Tudor-SN (middle panel) or seed-sequence-specific targeting mechanisms such as the recruitment of the ubiquitin ligase EBAX-1(right panel). See text for further details.

**Figure 2 ijms-24-06963-f002:**
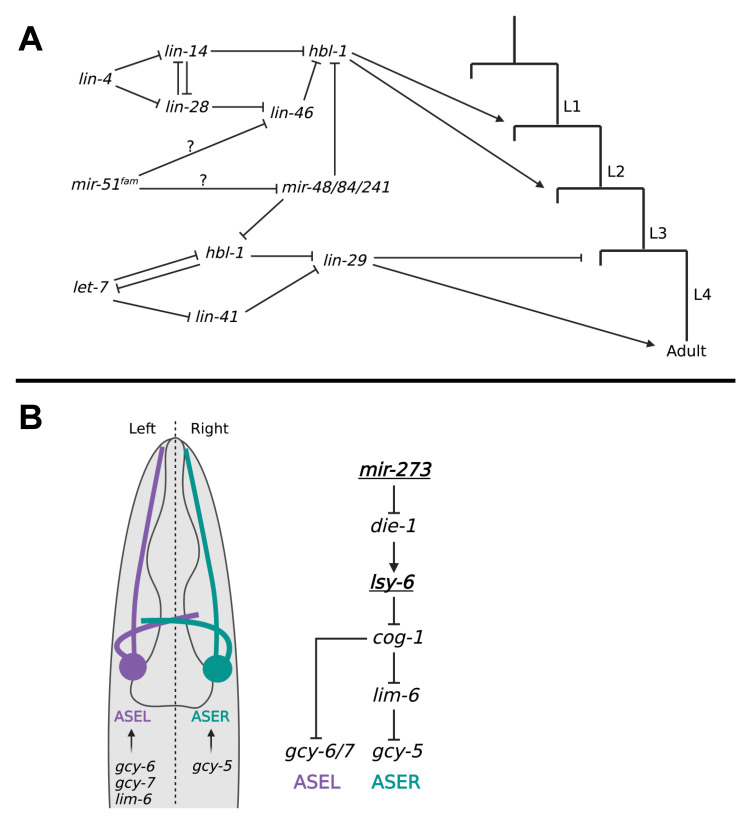
miRNA-mediated regulatory networks during *C. elegans* larval development. (**A**) The heterochronic pathway: *lin-4* and *let-7* coordinate cell lineage changes at each larval molt through the repression of several targets. *Lin-4* coordinates L1 and L2 cell lineages by repression of *lin-14* and *lin-28* which in-turn promotes expression of *hbl-1*. *Let-7* coordinates L3, L4 and adult lineages through the repression of *lin-41* and *hbl-1* which promotes the expression of *lin-29.* It is not yet clear how *mir-51^fam^* is involved in this pathway (question marks). Adapted from Refs. [107,108]. (**B**) Two miRNAs, *mir-273* and *lsy-6* (underlined), control ASEL/R cell identity by regulating the expression of *gcy* genes required for one cell fate or the other. Expression of *lsy-6* promotes the ASEL fate by promoting the expression of *gcy-6* and *gcy-7*, while expression of *mir-273* suppresses *lsy-6* and leads to *gcy-5* expression and the adoption of the ASER cell fate. Adapted from Refs. [109,110,111].

**Figure 3 ijms-24-06963-f003:**
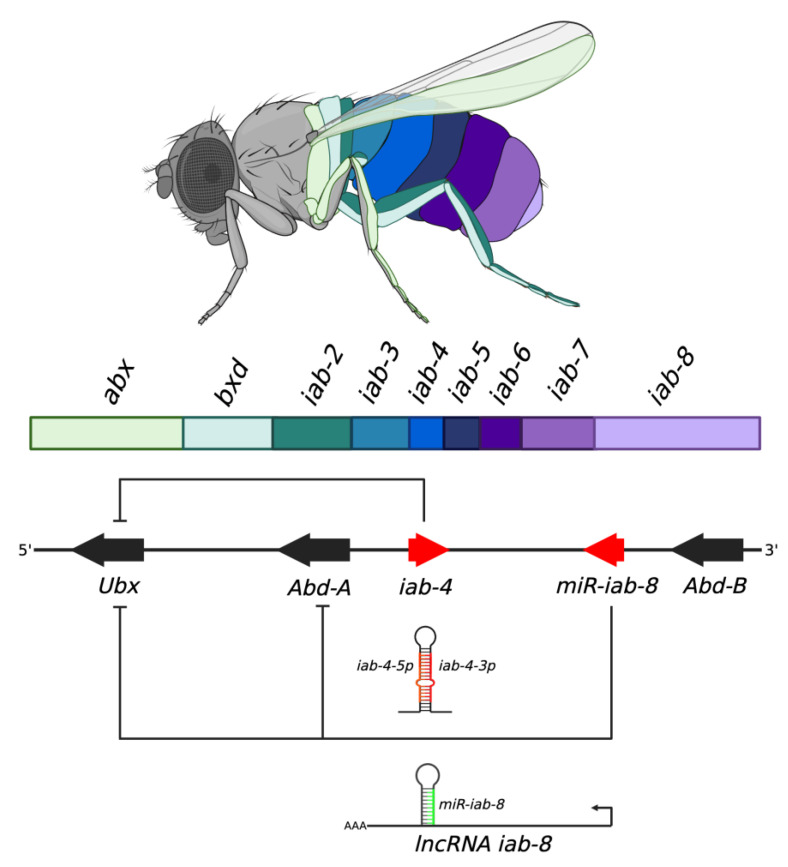
The bithorax complex in *Drosophila*. The nine cis-regulatory regions of the BX-C are shown in order (*abx, bxd, iab-2–iab8*) and color-coded to the regions they regulate in the developing fly. The homeotic genes *Ubx*, *Abd-A,* and *Abd-B* are shown in their respective positions in the cluster. The miRNAs *iab-4* and *miR-iab-8* are indicated by the red arrows. The *iab-4* miRNAs regulate *Ubx,* and *miR-iab*-8 regulates *Ubx* and *Abd-A*. The *iab-4* hairpin containing both the *iab-4-5p* and *iab-4-3p* miRNAs (red) and the lncRNA *iab-8* containing the *miR-iab-8* miRNA hairpin (green) are shown. BX-C DNA labeled 5′-3′ based on the orientation of the sense strand. Adapted from Refs. [136,140].

**Figure 4 ijms-24-06963-f004:**
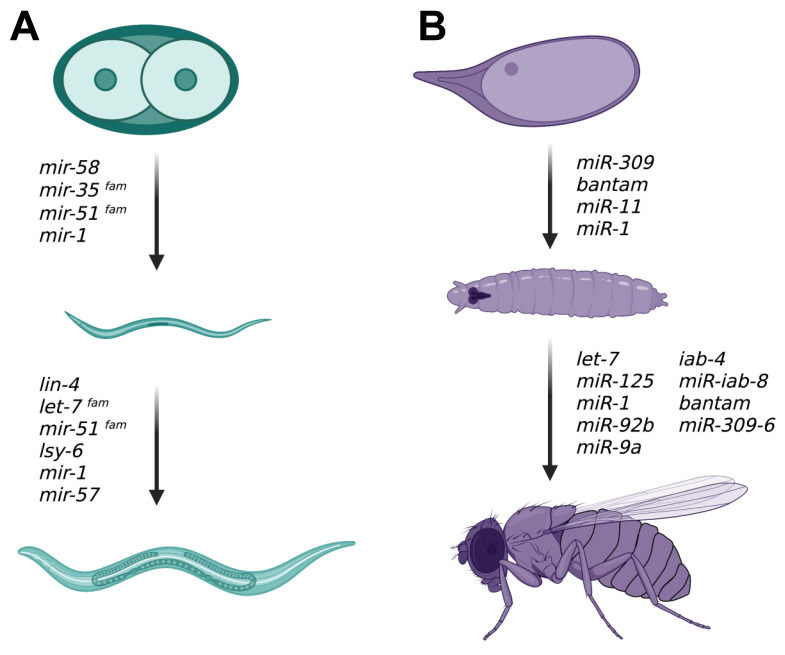
miRNAs involved in (**A**) *C. elegans* and (**B**) *Drosophila* development. Some of the many miRNAs that are required during both embryonic and larval development are listed for each invertebrate model organism.

## Data Availability

Not applicable.

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
