# Peer review of "A Compilation of the Diverse miRNA Functions in Caenorhabditis elegans and Drosophila melanogaster Development"

_ijms, 2023, doi:10.3390/ijms24086963_

Round 1

Reviewer 1 Report

The manuscript by Quesnelle at al. provides an overview of the roles of a subset of miRNAs involved in the development of two model invertebrate species, the nematode C. elegans and the fly D. melanogaster. Overall, I thought that the review provided some useful insights into the roles of miRNAs in some key aspects of invertebrate development, ranging from embryonic and larval development to tissue and pattern formation. In comparison to other reviews of miRNA function, many of which provide only a broad overview of miRNA functions, I found this article provides a more thorough description of some specific miRNAs’ functions, and thus, I think this review offers researchers a useful resource of miRNA function and some intriguing comparisons between two distantly related species.

I have only a few comments to make and will let the authors consider whether to incorporate any modifications suggested:

1. The main issue I had with this review was that it tended to bounce back and forth between C. elegans and D. melanogaster examples of miRNAs, and while useful comparisons were often made, sometimes the comparisons or connections of the roles of miRNAs in the two species were missing. Section 5.2 (Muscle Development) was one such section, where the narrative of the miRs in two species lacked any connection, either in miRs or common patterns of usage of them. Can the authors consider editing this section to provide some more effective comparisons?

2. Worms lack appendages, and hence Section 5.4 (Developing appendages) focused on the fly model, but what was missing were some comments on whether the worms have lost those miRs or if they have assumed different functions. Can the authors provide any insights on what types of miRs in general have retained similar roles and which have been lost or assumed new roles in these two divergent species?

3. The heterochronic roles of some miRNAs was nicely described, in terms of what genes are regulated by these miRNAs and how that gene regulation leads to properly timed development. What is known about the regulation of the miRNAs themselves? Clearly, they need to be expressed at the correct times and quantities. Can the authors provide any insights on how this is achieved?

4. Figure 2B is not mentioned in the text.

Author Response

Reviewer Report 1:

The manuscript by Quesnelle at al. provides an overview of the roles of a subset of miRNAs involved in the development of two model invertebrate species, the nematode C. elegans and the fly D. melanogaster. Overall, I thought that the review provided some useful insights into the roles of miRNAs in some key aspects of invertebrate development, ranging from embryonic and larval development to tissue and pattern formation. In comparison to other reviews of miRNA function, many of which provide only a broad overview of miRNA functions, I found this article provides a more thorough description of some specific miRNAs’ functions, and thus, I think this review offers researchers a useful resource of miRNA function and some intriguing comparisons between two distantly related species.

I have only a few comments to make and will let the authors consider whether to incorporate any modifications suggested:

  1. The main issue I had with this review was that it tended to bounce back and forth between C. elegans and D. melanogaster examples of miRNAs, and while useful comparisons were often made, sometimes the comparisons or connections of the roles of miRNAs in the two species were missing. Section 5.2 (Muscle Development) was one such section, where the narrative of the miRs in two species lacked any connection, either in miRs or common patterns of usage of them. Can the authors consider editing this section to provide some more effective comparisons?

Our review was not meant to say that the miRNAs work the same way in development for each of the two organisms, but rather we wanted to highlight the similarities and differences. We have clarified the section on muscle development - they work in a similar way and have a similar target muscle genes eg. Mef2

  1. Worms lack appendages, and hence Section 5.4 (Developing appendages) focused on the fly model, but what was missing were some comments on whether the worms have lost those miRs or if they have assumed different functions. Can the authors provide any insights on what types of miRs in general have retained similar roles and which have been lost or assumed new roles in these two divergent species?

While worms do not have appendages and segments like a fly, the worm does display A-P gene expression, for example the Hox gene nob-1 in C. elegans encodes a homolog of the Abd-B, and the nob-1 gene is regulated by an miRNA (mir-57).  Thus, we see that some of the miRNAs required in the developing fly appendages have gained new roles in the worm while some are not expressed in the worm at all. We have added a paragraph in this section that should help clarify these differences.

  1. The heterochronic roles of some miRNAs was nicely described, in terms of what genes are regulated by these miRNAs and how that gene regulation leads to properly timed development. What is known about the regulation of the miRNAs themselves? 

This is a very important question and is under active research, however, little is known.  One miRNA (let-7) has been well studied.  HBL-1, when expressed, inhibits let-7 in a negative feedback loop, which in itself prevents let-7 from being precociously expressed and we have mentioned this in the review.

Clearly, they need to be expressed at the correct times and quantities. Can the authors provide any insights on how this is achieved?

It is a good point, and it is like the chicken or egg problem, and unfortunately not much is known.

  1. Figure 2B is not mentioned in the text.

We have now mentioned Figure 2B in the text.

Reviewer 2 Report

The review article entitled “A compilation of the diverse miRNA functions in Caenorhabditis elegans and Drosophila melanogaster development” by Quesnelle et al highlight the function of different function of miRNA in development of small organism C. elegans and D. melanogaster. Authors summarized gene regulation by miRNAs in embryonic and larval development in both the organisms. Article is written very well and good for the general audience.

I have some suggestion on the article to improve the quality of the article:

1.     Authors can draw figures on the timeline of the milestone discoveries on miRNAs.

2.     A table could be added in the review of miRNAs involved in development, differentiations, and pathophysiological condition in both the organism for better understating.

3.     List of human disease associated miRNAs with orthologs in small organisms could be added in the review.

Author Response

Reviewer report 2:

The review article entitled “A compilation of the diverse miRNA functions in Caenorhabditis elegans and Drosophila melanogaster development” by Quesnelle et al highlight the function of different function of miRNA in development of small organism C. elegans and D. melanogaster. Authors summarized gene regulation by miRNAs in embryonic and larval development in both the organisms. Article is written very well and good for the general audience.

I have some suggestion on the article to improve the quality of the article:

  1. Authors can draw figures on the timeline of the milestone discoveries on miRNAs.

We don’t think there is enough milestones to warrant a timeline figure of its own, but instead we have indicated in the text when the major discovery was made.   Also, we have provided references for these milestones.

  1. A table could be added in the review of miRNAs involved in development, differentiations, and pathophysiological condition in both the organism for better understating.

We feel we don’t need to have a table that lists the miRNAs. This is essentially what is conveyed on Figure 4 of the review where we summarize all the miRNA discussed in the review and where they act during development. 

  1. List of human disease associated miRNAs with orthologs in small organisms could be added in the review.

This is a rapidly growing field and could be an entire review on its own, but we have mentioned miRNAs (families) that are conserved in worms and flies and humans and understanding their roles in development may contribute to our understanding of human diseases such as cancer.